# Pre-Emptive Drug Safety Evaluation of Iclepertin (BI-425809) Using Real-World Data and Virtual Addition of This Medication to the Actual Drug Regimen of Individuals from Large Populations

**DOI:** 10.3390/ph18101453

**Published:** 2025-09-28

**Authors:** Sebastian Härtter, Veronique Michaud, Matt K. Smith, Pamela Dow, Gerald Condon, Michael Desch, Jacques Turgeon

**Affiliations:** 1Translational Medicine & Clinical Pharmacology, Boehringer Ingelheim Pharma GmbH & Co, KG, Binger Str. 173, 55216 Ingelheim am Rhein, Germany; sebastian.haertter@boehringer-ingelheim.com (S.H.);; 2GalenusRx Inc., 17101 Porter Avenue, P.O. Box 560025, Montverde, FL 34756, USA; vmichaud@galenusrx.com (V.M.); pdow@galenusrx.com (P.D.); gcondon@galenusrx.com (G.C.)

**Keywords:** biosimulation, drug safety, CYP450, drug-induced LQTS, pharmacovigilance, real-world data, drug claims, iclepertin, BI-425809

## Abstract

**Introduction**. Adverse drug events (ADEs) are between the third and sixth most common cause of death worldwide. Biosimulation studies performed using real-world data could generate relevant drug safety information without exposing patients to ADEs. **Methods**. Iclepertin (BI-425809) was virtually added to the actual drug regimens of *n* = 4,405,063 individuals. Changes in risk level were estimated for drug-induced long QT syndrome and CYP450 drug interactions. The properties used for iclepertin included: dose of 10 mg (oral) once daily; bioavailability (F) = 71%; Cmax of 222 nM; CYP3A4 weak affinity substrate (partial metabolic clearance of ~80%); IC_50_ for hERG block of 30 μM. **Results**. A change in total medication risk score (MRS) was observed (6.3 ± 6.6 to 7.2 ± 6.6) following the addition of iclepertin in ~50% (*n* = 2,138,247) of the studied population. Among individuals with a change in MRS, ~65% had a 2-unit increase (max 11 units). The number of individuals classified in the High/Severe MRS category increased by 0.33%. The addition of iclepertin to individuals receiving CYP3A4 perpetrator drugs produced a greater change in MRS (+1.5) when compared to individuals not exposed to CYP3A4 perpetrators (+0.8). An additional 0.0032% of the population (*n* = 139) would be at risk of QT prolongation following the intake of iclepertin. Subset analyses performed in individuals with schizophrenia (targeted indication) demonstrated that these individuals had higher MRS values (13.0 ± 10.3) compared to those without schizophrenia (6.2 ± 6.9). However, the addition of iclepertin did not produce a greater increase in MRS in the schizophrenia population vs. the control population. Our pharmacoeconomic model did not account for any beneficial effects of the drug but the model based on MRS changes predicted a USD 91 yearly increase in medical expenditures (emergency department visits and hospitalizations) per individual (USD 3172 to USD 3263) following the addition of iclepertin. A similar increase was observed in the schizophrenia population following iclepertin addition. **Conclusions**. The increase in MRS associated with the addition of iclepertin to the drug regimen of a large population was minimal and mostly driven by CYP3A4 interactions. Using this model, interactions can be identified a priori, making risk mitigable and preventable without exposing patients to toxicity.

## 1. Introduction

Scientists, clinicians, and regulatory agencies have long recognized that adverse drug events (ADEs) are between the third and sixth most common cause of death worldwide [1,2,3,4]. The association of ADEs with substantial morbidity and mortality has resulted in mandatory phase IV clinical trials and black box warnings and the withdrawal of drugs from the market [5]. Drug safety monitoring through pharmacovigilance studies must remain in effect after the approval and marketing of medications [5,6]. Drug safety monitoring is especially critical for side effects that are rare (e.g., drug-induced torsade de pointes), occur in certain sub-groups of the population (e.g., genetic predisposition due to drug metabolism or drug transporters), or recognized long after drug approval (e.g., thalidomide and congenital malformations) [7,8,9].

On one hand, (i) published case reports, (ii) Medwatch (since 1993) linked to the FDA Adverse Drug Event Reporting System (FAERS), (iii) the emergence of information technology such as electronic health records (EHRs) since the 2009 Health Information Technology for Economic and Clinical Health (HITECH) Act, (iv) the Sentinel initiative, (v) data partnership networks, and (vi) consortia, such as PedsNet and the Open Health Data Science Informatics (OHDSI) network, have increased the capability to capture information about drug safety [5]. On the other hand, some consider these resources antiquated and believe that applied statistical signal detection methodologies have limited usefulness [1]. A major limitation of all of these listed approaches is that they require that events occur before they can be captured and analyzed, at which point safety-related warnings are added and negative reports are generated. In other words, some patients must experience side effects and potentially mortality before any safety-related changes occur. For new market entities, some safety label changes occur 10 years after drug approval [10,11]. Therefore, continued surveillance through a drug’s lifecycle after approval is critical to trigger safety-related drug label changes [10,11].

One such monitoring solution for drug safety is a medication-based risk score, which can be used to identify at-risk individuals who would benefit from medication management interventions, as well as reduce inappropriate polypharmacy, adverse health outcomes, and avoidable healthcare utilization [2,12,13]. Various medication risk scoring systems, like indices, have been developed to quantify the complexity of an individual’s medication regimen, risk of medication-related falls, sedative load, anticholinergic burden, or drug regimen appropriateness [12,14,15]. In the last 25 years, the authors have contributed to the development and evolution of clinical decision support systems (CDSSs) [16], where each CDSS included a drug regimen risk score that could be used to foresee patient outcomes, such as ADEs, falls, hospitalization, emergency department (ED) visits, and death [17,18,19,20]. The authors have shown that medication-based risk scores influence pharmacists’ interventions and affect patients’ outcomes [21,22].

On 31 December 2019, the World Health Organization issued an epidemiological alert in response to an unidentified pneumonia in Wuhan, China [23,24]. Rapidly, SARS-CoV-2 spread globally, causing the coronavirus disease 2019 (COVID-19) pandemic. The scientific community sought potential treatments using repurposed drugs such as hydroxychloroquine, chloroquine (alone or in combination with azithromycin), lopinavir/ritonavir, ivermectin, ebselen, remdesivir, molnupivir, favipiravir, bebtelovimab, sotrovimab, and crizanlizumab, among others [25]. Considering the known toxicity of some of these agents and their potential for drug–drug interactions, a new “pre-emptive pharmacovigilance” strategy was developed [26]. This strategy was used to design polypharmacy biosimulation studies where repurposed drugs were virtually added to the real drug regimens of individuals and changes in medication risk scores were monitored to identify potential ADEs and negative outcomes, including an increased risk of hospitalization, ED visits, and medical expenditures [27,28].

A similar strategy is applied to our study reported herein, where a new chemical entity, iclepertin (BI-425809), was virtually added to the drug regimens of millions of individuals. As iclepertin is intended to be used in patients with schizophrenia, a subset analysis was performed in this population. Risk assessments of drug-induced torsade de pointes and CYP450 drug–drug interaction burden were the focus of our study.

## 2. Results

Data were obtained for *n* = 4,435,330 individuals. Based on the criteria listed above, the following numbers of individuals were retained in each group: commercially insured = 1,937,389; Medicaid = 1,983,976; and Medicare = 483,698; of those, *n* = 4,405,063 were retained for analyses. Baseline characteristics for the total population and across the Commercial, Medicaid, and Medicare individuals are presented in Table 1. There was a larger proportion of women in the Medicaid group. Further, Medicare beneficiaries were significantly older when compared to the Commercial and Medicaid groups. Medicare beneficiaries also had the highest Charlson Comorbidity Index (CCI), followed by the Medicaid and Commercial populations. Medicaid individuals took more drugs on average than Medicare beneficiaries and both groups were taking more medications than Commercial individuals. The 60–69 years of age group in the Commercial and Medicaid populations were taking more drugs than their younger counterparts (Table 2). The top 50 prescribed medications per coverage group are reported in Appendix A.

Individuals in the Medicaid population had a higher number of ED visits than the two other groups (Table 1). Further, hospitalizations were more common in the Medicaid and Medicare groups, while hospital length of stay (LOS) showed a similar distribution among the three groups. Falls were more frequent in Medicare beneficiaries. The MRS was higher in Medicaid individuals, followed by Medicare beneficiaries, and more Medicaid individuals were in the Severe category when compared to the two other groups (Figure 1). Medicaid beneficiaries also showed a higher risk for CYP450 drug interactions and an increased risk for drug-induced long QT syndrome (LQTS; Table 1).

Table 3 reports our findings following the addition of iclepertin to the drug regimen of individuals from the different coverage groups. As no information was available for iclepertin in the FAERS, the combined effect of this parameter on the MRS could not be estimated. Clinical data for the anticholinergic properties and sedative load characteristics of iclepertin were also not available. A change in anticholinergic burden or sedative load is not expected based on iclepertin drug disposition characteristics (weak CYP3A4 substrate) and the low likelihood of iclepertin to cause a change in the exposure of other drugs. Note, the addition of iclepertin was associated with a slight increase (0.85 to 0.88 points; considered not significant) in the MRS on average. Nevertheless, an increase in the MRS was observed in about 50% of the population. Some individuals had an increase of up to 11 points; the various degrees of increase in the MRS led to changes in the distribution of individuals in each of the MRS categories (Figure 1). For all three groups, fewer individuals were in the “Minimal” category following the addition of iclepertin, and more individuals were distributed towards higher risk categories (Table 3, Figure 1).

More discrete analyses were performed for both the CYP450 drug interaction burden and LQTS score as these two factors were significantly modified by CYP3A4 inhibitors and strong CYP3A4 affinity substrates: as iclepertin is expected to exhibit a weak affinity towards CYP3A4, it would behave like a victim drug. Table 4 reports on the detailed analyses performed. The presence of CYP3A4 inhibitors and strong affinity substrates was associated with higher MRS at baseline in all populations tested. The list of clinically relevant perpetrator interacting drugs including CYP3A4 inhibitors, CYP3A4 inducers, and stronger affinity CYP3A4 competitive substrates are listed in Appendix A.

The addition of iclepertin increased the MRS by 0.8 points in the Medicare population, 1.6 points in the Medicaid population, and 1.9 points in the Commercial population. As mentioned previously, this increase was mostly explained by the CYP450 drug interaction burden parameter.

Similarly, the presence of concomitant CYP3A4 inhibitors and CYP3A4 strong affinity substrates was associated with a higher LQTS risk score at baseline. The addition of iclepertin did not significantly modify the LQTS score, except for in some specific individuals (Table 5). Overall, *n* = 139 individuals had an increase in their LQTS score of 2 points and above, 123 had an increase in the LQTS score of 3 points and above, and 113 had an increase in their LQTS score of 5 points and above. An increase of 5 points and above was mostly observed in the Medicaid population.

Iclepertin is a glycine transporter 1 (GlyT1) inhibitor intended to be used for the treatment of cognitive impairment associated with schizophrenia. A sub-analysis was performed in individuals with listed ICD-10 codes associated with schizophrenia (Table 6). *n* = 123,722 individuals met the required conditions (ICD-10 codes of F-20 to F-29). In all sub-groups (Commercial, Medicaid, or Medicare), schizophrenia was associated with a much higher MRS (6.8 points on average). Changes in MRS produced by the addition of iclepertin in the schizophrenia population were of the same magnitude as those observed in the control population. Further, individuals in the schizophrenia population had higher CYP450 drug interaction burden score and higher drug-induced LQTS score than the control population. Changes in these parameters produced by the addition of iclepertin were also of the same magnitude as changes observed in the control population.

A pharmacoeconomic evaluation of the effect of adding iclepertin to the actual drug regimens of individuals on medical expenditures, ED visits, and hospitalizations based on the computed increase in MRS is presented in Table 7. Overall, the increase in MRS (0.85 to 0.88) was associated with a postulated modest increase in average medical expenditure over the entire cohort (USD 91 on average). The model also predicted a 0.01% to 0.03% increase in ED visits and a 0.004% to 0.006% increase in hospitalizations. Table 8 reports the impact of CYP3A4 inhibitors and CYP3A4 strong affinity substrates on the increases in medical expenditures, ED visits, and hospitalizations.

The effect of perpetrator CYP3A4 inhibitors and CYP3A4 strong affinity substrates was observed both at baseline and after the addition of iclepertin. Table 9 reports on the postulated impact of iclepertin on medical expenditures, ED visits, and hospitalizations in individuals with schizophrenia. Changes observed were of the same magnitude as those observed in the non-schizophrenia population. However, the benefits associated with the use of iclepertin in individuals with schizophrenia were not considered in our pharmacoeconomic evaluation.

## 3. Discussion

In this study, we determined the safety element profile of a new chemical entity, iclepertin, by using real-world claims data and virtually adding iclepertin to the actual drug regimens of over 4 million individuals. Based on some pharmacokinetic (CYP3A4 weak affinity substrate with partial metabolic clearance of 80%) and pharmacodynamic (IC_50_ for block of I_Kr_) properties, we estimated how the virtual intake of this drug would expose subjects to significant multidrug interactions and potential side effects. Relevant side effect frequency for iclepertin is absent from the FAERS, and therefore, our approach did not estimate any benefits associated with iclepertin or consider other potential side effects. However, we demonstrated that the safety profile of iclepertin was similar in the targeted population of individuals with schizophrenia compared to the non-schizophrenia population. This approach represents a new, pre-emptive, polypharmacy biosimulation strategy that adds to the pharmacovigilance armamentarium; this science is proactive rather than reactive. More importantly, information is obtained without exposing any individuals to drugs and potential side effects, including death.

A change in MRS was observed following the addition of iclepertin in about 50% of the population (*n* = 2,138,247). The percentages of individuals with a change in MRS and the mean increases in MRS were similar between the three tested groups (47.3–50.6% change in MRS and percent increase of 0.85–0.88 units, respectively). The MRS was classified into Minimal, Low, Intermediate, High, and Severe categories that have been associated with health outcomes [18,19,22,29]. It was previously demonstrated that Intermediate and High/Severe categories are associated with increased risk of poor health outcomes, including ADEs, fall, death, and medical expenditures, compared to the Minimal category [18,19]. When looking at individuals with a change in their MRS category from the baseline after the addition of iclepertin, the simulation predicted that 3.33% of the population would have a change in MRS leading to a higher risk category. The number of individuals classified in the High/Severe MRS category increased by 0.33% in the overall population. By sub-group, an increase of *n* = 4070 (0.21%), 9201 (0.46%), and 1256 (0.26%) individuals was estimated in the High/Severe MRS category for the Commercial, Medicaid, and Medicare groups, respectively.

In this study, changes observed in MRS secondary to the virtual introduction of iclepertin to the drug regimen of commercially insured and Medicare or Medicaid beneficiaries were 3 to 10 times less than those observed in similar populations from our previous studies [28]. Previously, the virtual addition of drugs such as hydroxychloroquine or chloroquine (alone or with azithromycin) or lopinavir/ritonavir to the drug regimen of commercially insured and Medicare beneficiaries during the COVID-19 outbreak led to more significant increases in the MRS [28]. In the later study, changes in MRS were mostly due to an increase in CYP450 drug interaction and drug-induced LQTS risk indices. In another study, simulations performed in a group of participants in the Program for All-inclusive Care of the Elderly (PACE) with the same COVID-19 repurposed drugs led to similar results [27].

Based on available data, iclepertin was considered a weak affinity substrate for CYP3A4. Therefore, iclepertin systemic exposure can be affected if administered with CYP3A4 inhibitors, CYP3A4 inducers, or CYP3A4 higher affinity substrates (i.e., other drugs becoming competitive substrate inhibitors). This study looked at relevant CYP3A4 interactions, as most drugs are metabolized by CYP450 enzymes, driving the effect on drug interaction burden [30]. Across the three populations, atorvastatin was the most common clinically relevant concomitant CYP3A4 interacting drug, followed by omeprazole, amlodipine, simvastatin, buspirone, doxycycline, topiramate, buprenorphine, fluconazole, and risperidone. Following the addition of iclepertin, the CYP450 interaction burden score was estimated to increase by approximately one unit for the overall population (Table 1 and Table 3). The simulation indicates that a similar magnitude of CYP450 interaction burden score change is expected among the various groups tested.

Our model predicted a change in drug-induced LQTS score in 0.0032% of the population (*n* = 139). Overall, changes in the LQTS score were minimal. LQTS scores were classified into Low-, Moderate-, and High-risk groups for QT prolongation. Based on our model, the risk of experiencing torsade de pointes is associated with the High-risk group [26]. Following the addition of iclepertin, no individual who had an increase in their LQTS score became at high risk of having QT prolongation and torsade de pointes; *n* = 137 remained as low risk and two individuals moved from low to moderate risk. Notably, women are at increased risk of torsade de pointes [31,32], and the current simulation did not find a higher proportion of women in the increased LQTS score group. An increased risk for women was previously demonstrated in simulations studies of known QT prolongation drugs [27,28]. This suggests that QT prolongation risk of iclepertin is likely negligible.

Across all populations, the average MRS increased from 6.3 ± 6.6 to 7.2 + 6.6 following the virtual addition of iclepertin to individuals’ drug regimens. Further, 1,382,567 individuals had an MRS increase of 2 units, 72,506 individuals had an increase of 5 units, 375 individuals had an increase of 10 units, and 5 individuals had an increase of 11 units. A previously published model, trained using claims data, was used for the current simulation [18]. When examining medical expenditures, ED visits, and hospitalizations based on an increase in MRS following the virtual addition of iclepertin, the updated model predicted a USD 91 increase per individual (USD 3172 to USD 3263) for the total population. The Commercial population was predicted to have a USD 131 increase in cost, while the Medicaid and Medicare populations had a USD 62 and USD 137 predicted increase in cost, respectively. The pharmacoeconomic estimation performed is biased at this stage as it does not consider beneficial effects and potential savings associated with drug efficacy.

Iclepertin was developed as a potent and selective glycine 1 transporter (GlyT1) inhibitor to improve symptoms of cognitive impairment associated with schizophrenia [33]. Sub-analyses were conducted in individuals with a schizophrenia diagnosis based on ICD-10 codes to compare changes in MRS in this population versus the non-schizophrenia population. Our results demonstrated that changes in MRS, in CYP450 interaction burden, and drug-induced long QT syndrome indices did not differ between subjects with schizophrenia vs. without known schizophrenia. As we have observed that individuals with schizophrenia in the three populations tested had about a 2-fold increase in their MRS compared to the non-schizophrenia population, these findings are important.

Previously, Michaud et al. [18] explored the association of the MRS and health outcomes in a large Medicare population. Study results showed that a 1-unit change in the MRS was associated with an 8.5% increase in total medical expenditures (Part A and B); therefore, a 2-unit change in MRS would be associated with an increase of 17.7%, and a 10-unit change in MRS with 126% increase in medical costs [18]. Based on the current model developed for the total population (three groups), a 1-unit change in MRS was associated with an increase of 4% in medical costs, which would represent an increase in medical expenditures of 8.1% and 47.3% for individuals with a predicted increase of 2 and 10 units of the MRS, respectively. In the current simulation, the MRS was derived using the most recent drug claims in 2019 (with retroactive drug claim overlap), while the published model used the maximum MRS generated over the year [18]. Therefore, the calculated MRS and, consequently, the impact on medical expenditures may be underestimated for some individuals.

The previously developed model by Michaud et al. [18] found that per 1-unit increase of the MRS per year, the number of expected ED visits increased by 7%, and the number of hospital admissions was predicted to increase by 3%. The current model trained for iclepertin drug simulation estimated a 4.2% increase in ED visits and a 5.8% increase in hospital admission per 1-unit increase in MRS, comparatively. The previously published model estimated a 5.8% increase in odds ratio of having an ADE per increase in MRS unit. Michaud et al. [18] also reported that a 2-unit increase in the MRS (as estimated for 65% of individuals in this simulation) could translate into a 12% increase in odds ratio for experiencing at least one ADE. Therefore, an increase of up to 86% in the odds ratio of having an ADE is estimated for an increase of 11 units of the MRS, which is the highest increase in the MRS observed in this simulation.

As mentioned, relevant side effect frequency for iclepertin is absent from the FAERS, and therefore our approach did not estimate benefits associated with iclepertin or consider other potential side effects. This is a significant limitation as only two out of five factors included in the MRS could be assessed. Further, using real-world data presents several advantages, including the acquisition and analysis of data on many individuals in a short time. However, real-world data often requires significant clean-up, is static, and is not always uniform between all individuals, which can be a limitation when needing specific inclusion criteria to perform analyses.

Finally, the presence of CYP3A4 inhibitors or CYP3A4 strong affinity substrates was the most important parameter driving changes in medical expenditures, ED visits, and hospitalizations, both at baseline and following the addition of iclepertin; this observation remained true in the schizophrenia population.

## 4. Materials and Methods

### 4.1. Dataset and Study Population

The biosimulation study was conducted using the Merative™ MarketScan^®^ Research Databases (Merative, Francisco Partners, 100 Phoenix Dr., Ann Arbor, MI, USA). Dataset requirements were as follows:(1)Randomly selected individuals with at least one drug claim from 1 January 2019, through 31 December 2019. The year 2019 was selected to avoid bias introduced by the outstanding conditions associated with the COVID-19 pandemic.(2)Individuals must be adults (≥18 years).(3)Individuals must be either commercially insured or eligible for Medicare or Medicaid. Commercially insured individuals are those obtaining insurance either through an employer or family plan. Medicaid is a wide-ranging, joint federal and state healthcare program mostly for low-income individuals of any age. Medicare is a federal program which provides healthcare coverage to individuals 65 years of age and older. Alive individuals had to be enrolled for the entire year of 2019.(4)Tables and variables (A: annual summary enrollment; B: outpatient drug claims; F: facility header; I: inpatient admission; L: long-term care (Medicaid); O: outpatient services; R: lab data; S: inpatient services; T: detail enrollment) were obtained for the full year of 2019 for selected individuals.

Drugs found to have no specific “day supply data”, but for which claims data indicated some refills, were analyzed and assigned as a “chronic” or “acute” medication based on their clinical indication. If drugs were considered as chronic medication (e.g., opioids, antihypertensive agents (beta-blockers, calcium channel blockers), statins, metformin, sulfonylureas), the drug was considered continuous for the full year (from this date) for the risk stratification analysis. If the medications were considered acute (e.g., antibiotics, seasonal allergy medications), the drug was considered administered for 30 days (from the date prescribed).

ICD-10 codes F20 to F29 were used to identify individuals with schizophrenia.

### 4.2. hERG IC50 Determination

hERG IC_50_ was determined by patch-clamp in HEK293 cells, as described previously [34].

### 4.3. Medication Risk Score

The medication risk score (MRS) used for our analyses was previously described in detail [27,28,29,35]. Briefly, it is derived from aggregated, weighted values for five pharmacokinetic and pharmacodynamic factors shown to be associated with medication-related morbidity and mortality [18,19,29]. The total MRS ranges from 0 to 53. Algorithms used for LQTS index determination were previously described in Turgeon et al. [36].

### 4.4. Data Properties

We used the following properties for the simulations with iclepertin: an oral dose of 10 mg once daily, an absolute bioavailability (F) of 71%, a maximum plasma concentration (Cmax) after single dose of 222 nM, an estimated weak affinity substrate of CYP3A4, with a partial metabolic clearance through CYP3A4 approximated to 80%, and an IC_50_ value for the block of the rapid component of the delayed rectifier potassium current (I_Kr_; human ether-a-go-go protein (hERG)) estimated at 30 μM.

### 4.5. Data Analysis

To simulate the effects of adding iclepertin on the MRS, a fictitious claim was created for each individual in the database by adding this drug to their actual drug regimen. Then, a new MRS was derived for each individual. To perform the medication risk stratification, a webservice interface and customized scripts were used. MRSs were generated by processing prescribed drug claims using national drug codes (NDCs) as drug identifiers. Medication data were extracted from the claims data and cleaned of errors and inconsistencies through quality and integrity analyses. This included identifying claims that have been reversed and removing them from the dataset, estimating days of supply for drug claims (described previously), and identifying and removing any vaccine claims from the dataset. As NDCs can also denote non-medications (e.g., medical devices, medical supplies), active medication data was further filtered to exclude such NDCs. Active medication data for each individual was filtered based on prescription dates and days of supply and included any refills. Finally, the Charlson Comorbidity Index (CCI), calculated from the sum of scores for each comorbid condition listed in the Charlson Comorbidity Score Mapping Table, was used to account for multimorbidity and compare the disease severity in each group [37]. Descriptive population characteristics and individual risk factors, including means, medians, standard deviations, ranges, and proportions, were measured, as appropriate. Analyses were performed in Python, SciPy, Statsmodels, and Matplotlib. Microsoft SQL Server was used to manipulate and analyze large datasets.

### 4.6. Statistical Analyses

Given the deterministic nature of this study, statistical significance analysis is not warranted. Exactly one fictitious claim was added to all patients’ drug regimens, which were analyzed using the deterministic MRS algorithm. Therefore, there is no null hypothesis with which to compare and common sense should be used to determine if differences observed are meaningful and clinically relevant [38]. For factors associated with changes in the MRS and its components, a difference of at least 20% was considered significant.

## 5. Conclusions

Results suggest that the increase in MRS was mostly affected by CYP3A4 potential multidrug interactions, with an increase in medical expenditures, ED visits, and hospitalizations, both at baseline and following the addition of iclepertin. The mean MRS increase remains relatively low for the overall population (0.8 unit). However, among those who had a change in MRS, ~65% had an increase of 2 units, with some individuals experiencing an increase up to 11 units. The most prescribed drugs that may contribute to multidrug interactions have also been identified. Hence, interactions and potential CYP3A4 perpetrators can be identified a priori, and the risk can be mitigated and prevented if appropriate recommendations are proposed. Finally, at the dose tested, the risk of QT prolongation associated with iclepertin is present in a very small percentage of the population. The proposed simulation strategy can help identify individuals at increased risk of ADEs and poorer health outcomes.

## Figures and Tables

**Figure 1 pharmaceuticals-18-01453-f001:**
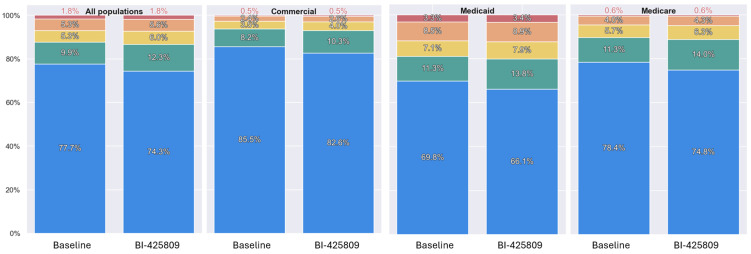
Medication Risk Score (MRS) distribution at baseline and after the addition of iclepertin by population and MRS categories. MRS Categories: Minimal (Blue), Low (Green), Intermediate (Yellow), High (Orange), Severe (Red). Although changes in MRS distribution were observed for each group, the extent of changes was of relatively minor, suggesting a good safety profile of iclepertin.

**Table 1 pharmaceuticals-18-01453-t001:** Population characteristics at baseline before the virtual addition of iclepertin.

Individuals’ Characteristics	Total Population	Commercial	Medicaid	Medicare
Number of individuals	4,405,063	1,937,389	1,983,976	483,698
Number of females: *n* (%)	2,720,831 (61.8%)	1,101,198 (56.8%)	1,346,377 (67.9%)	273,256 (56.5%)
Age, years: mean ± SD (range)	46.7 ± 17.2 (19–89)	43.8 ± 13.0 (19–64)	42.7 ± 16.32 (19–89)	74.6 ± 7.12 (20–89)
<60 years: *n* (%)	3,333,851 (75.8%)	1,696,396 (87.6%)	1,635,910 (82.5%)	1545 (0.3%)
60–69 years: *n* (%)	615,989 (14.0%)	240,993 (12.4%)	237,336 (12.0%)	137,660 (28.5%)
70–79 years: *n* (%)	288,798 (6.6%)		64,997 (3.3%)	223,801 (46.3%)
≥80 years: *n* (%)	166,425 (3.8%)		45,733 (2.3%)	120,692 (25.0%)
CCI: mean ± SD (range)	1.2 ± 2.2 (0–24)	0.7 ± 1.6 (0–22)	1.5 ± 2.5 (0–24)	2.0 ± 2.7 (0–22)
0: *n* (%)	2,660,993 (60.4%	1,396,289 (72.1%)	1,086,499 (54.8%)	178,205 (36.8%)
1–2: *n* (%)	1,095,546 (24.9%)	400,238 (20.7%)	520,512 (26.2%)	174,796 (36.1%)
3–4: *n* (%)	259,553 (5.9%)	46,339 (2.4%)	145,688 (7.3%)	67,526 (14.0%)
≥5: *n* (%)	388,971 (8.8%)	94,523 (4.9%)	231,277 (11.7%)	63,171 (13.1%)
Number of medications per individual: mean ± SD+D (range)	3.9 ± 3.8 (1–98)	3.2 ± 2.6 (1–76)	4.6 ± 4.7 (1–98)	3.9 ± 3.1 (1–36)
Number of ED visits per individual per year: mean ± SD (range)	0.9 ± 2.3 (0–339)	0.3 ± 0.9 (0–187)	1.6 ± 3.2 (0–339)	0.5 ± 1.1 (0–72)
#1 ED visit: *n* (%)	721,553 (16.4%)	230,149 (11.9%)	418,596 (21.1%)	72,808 (15.1%)
#2 ED visits: *n* (%)	324,667 (7.4%)	62,470 (3.2%)	235,323 (11.9%)	26,874 (5.6%)
≥#3 ED visits: *n* (%)	454,329 (10.3%)	38,679 (2.0%)	393,046 (19.8%)	22,604 (4.7%)
Number of hospitalizations per individual per year: mean ± SD (range)	0.1 ± 0.6 (0–57)	0.1 ± 0.3 (0–24)	0.2 ± 0.9 (0–57)	0.2 ± 0.6 (0–20)
#1 hospitalization: *n* (%)	258,866 (5.9%)	55,593 (2.9%)	155,680 (7.8%)	47,593 (9.8%)
#2 hospitalizations: *n* (%)	71,328 (1.6%)	10,533 (0.5%)	50,150 (2.5%)	10,638 (2.2%)
≥#3 hospitalizations: *n* (%)	51,644 (1.2%)	5336 (0.3%)	41,390 (2.1%)	4918 (1.0%)
LOS per visit per individual (days): mean ± SD (range) *	5.9 ± 6.5 (1–481)	5.0 ± 5.3 (1–469)	6.3 ± 7.2 (1–481)	5.1 ± 4.4 (1–235)
1–7 days: *n* (%)	306,649 (80.3%)	60,988 (85.3%)	192,895 (78.0%)	52,766 (83.6%)
8–14 days: *n* (%)	57,215 (15.0%)	76,195 (10.7%)	41,178 (16.7%)	8418 (13.3%)
15–21 days: *n* (%)	11,018 (2.9%)	1753 (2.5%)	7910 (3.2%)	1355 (2.1%)
≥22 days: *n* (%)	6956 (1.8%)	1102 (1.5%)	5244 (2.1%)	610 (1.0%)
Number of individuals with ≥1 ADE: *n* (%)	335,808 (7.6%)	64,058 (3.3%)	246,440 (12.4%)	25,310 (5.2%)
Number of individuals with ≥1 fall: *n* (%)	148,160 (3.4%)	21,975 (1.1%)	95,538 (4.8%)	30,647 (6.3%)
MRS: mean ± SD (range)	6.4 ± 7.1 (0–47)	4.9 ± 5.3 (0–43)	8.0 ± 8.4 (0–47)	6.1 ± 6.1 (0–44)
MRS category				
Minimal: *n* (%)	3,420,644 (77.7%)	1,656,527 (85.5%)	1,385,093 (69.8%)	379,024 (78.4%)
Low: *n* (%)	437,208 (9.9%)	159,052 (8.2%)	223,556 (11.3%)	54,600 (11.3%)
Intermediate: *n* (%)	235,514 (5.3%)	67,125 (3.5%)	140,772 (7.1%)	27,617 (5.7%)
High: *n* (%)	233,655 (5.3%)	45,712 (2.4%)	168,386 (8.5%)	19,557 (4.0%)
Severe: *n* (%)	78,042 (1.8%)	8973 (0.5%)	66,169 (3.3%)	2900 (0.6%)
CYP450 drug interaction burden: mean ± SD (range)	1.6 ± 2.8 (0–20)	1.2 ± 2.2 (0–20)	2.1 ± 3.3 (0–20)	1.6 ± 2.5 (0–20)
LQTS score: mean ± SD	0.4 ± 1.4	0.2 ± 0.9	0.6 ± 1.8	0.4 ± 1.3

Abbreviations: ADE—Adverse Drug Event; CCI—Charlson Comorbidity Index; CYP450—Cytochrome P450; ED—Emergency Department; LQTS—Long QT Syndrome; LOS—Length of Stay; MRS—Medication Risk Score; SD—Standard Deviation. * LOS is defined for individuals who are admitted and hospitalized. For individuals who are admitted and discharged on the same day, it is considered as a one-day stay.

**Table 2 pharmaceuticals-18-01453-t002:** Number of drugs per individual by age group (mean ± SD).

Age	Total Population	Commercial	Medicaid	Medicare
Total population	3.9 ± 3.8	3.2 ± 2.6	4.6 ± 4.7	3.9 ± 3.1
<60 years	3.7 ± 3.6	3.1 ± 2.5	4.3 ± 4.4	4.2 ± 4.0
60–69 years	5.1 ± 4.5	4.3 ± 3.2	6.6 ± 5.7	3.7 ± 3.0
70–79 years	4.2 ± 3.8	NA	5.4 ± 5.4	3.9 ± 3.1
≥80 years	4.3 ± 3.2	NA	4.5 ± 4.6	4.2 ± 3.2

Abbreviations: NA—Not Applicable.

**Table 3 pharmaceuticals-18-01453-t003:** Population characteristics following the virtual addition of iclepertin to the actual drug regimen of the individuals.

Individuals’ Characteristics	Total Population	Commercial	Medicaid	Medicare
Number of individuals	4,405,063	1,937,389	1,983,976	483,698
Number of medications per individual: mean ± SD + D (range)	4.9 ± 3.5 (2–99)	4.2 ± 2.6 (2–77)	5.6 ± 4.7 (2–99)	4.9 ± 3.1 (2–37)
MRS: mean ± SD (range)	7.3 ± 7.0 (0–47)	5.8 ± 5.4 (0–43)	8.8 ± 8.2 (0–47)	7.0 ± 6.1 (0–44)
Delta vs. baseline:				
mean; maximum	0.87; 11	0.88; 11	0.85; 10	0.85; 10
Number of individuals with a change in MRS: *n* (%)	2,138,247 (48.5%)	955,515 (49.3%)	937,515 (49.3%)	244,792 (50.6%)
MRS bin: change in number of individuals from baseline				
Minimal: *n* (%)	−146,610 (−3.33%)	−55,593 (−2.86%)	−73,860 (−3.72%)	−17,157 (−3.55%)
Low: *n* (%)	+103,204 (+2.34%)	+41,002 (+2.12%)	+49,265 (+2.48%)	+12,937 (+2.67%)
Intermediate: *n* (%)	+28,879 (+0.66%)	+10,521 (+0.54%)	+15,394 (+0.78%)	+2964 (+0.61%)
High: *n* (%)	+13,255 (+0.30%)	+3854 (+0.20%)	+8214 (+0.41%)	+1187 (+0.25%)
Severe: *n* (%)	+1272 (+0.03%)	+216 (+0.01%)	+987 (+0.05%)	+69 (0.01%)
CYP450 interaction burden: mean ± SD (range)	2.5 ± 2.7 (0–20)	2.0 ± 2.3 (0–20)	3.0 ± 3.1 (0–20)	2.4 ± 2.5 (0–20)
Delta vs. baseline: mean; maximum	0.87; 10	0.88; 10	0.88; 10	0.85; 10
LQTS score: mean ± SD	0.4 ± 1.4	0.2 ± 0.9	0.6 ± 1.8	0.4 ± 1.3
Delta vs. baseline: mean; maximum	0.0001; 5	0.0001; 5	0.0002; 5	0.0003; 5

Abbreviations: CYP450—Cytochrome P450; LQTS—Long QT Syndrome; MRS—Medication Risk Score; SD—Standard Deviation.

**Table 4 pharmaceuticals-18-01453-t004:** Medication Risk Score (MRS), CYP450 drug interaction burden (CIB), and LQTS score in individuals receiving CYP3A4 perpetrators drugs.

Groups	Total Population ^¥^	Commercial ^¥^	Medicaid ^¥^	Medicare ^¥^
MRS				
Baseline MRS: Mean ± SD				
Control *	6.2 ± 6.9	4.7 ± 5.3	7.7 ± 8.2	5.7 ± 5.8
CYP3A4 inhibitors/strong affinity substrates	11.9 ± 9.6	9.3 ± 8.1	13.6 ± 10.8	12.2 ± 7.2
Simulation with iclepertin; MRS: Mean ± SD				
Control	7.0 ± 6.9	5.6 ± 5.3	8.5 ± 8.1	6.5 ± 5.8
CYP3A4 inhibitors/strong affinity substrates	13.4 ± 8.3	11.2 ± 6.7	15.2 ± 9.3	13.0 ± 6.6
CIB				
Baseline CIB: Mean ± SD				
Control	1.5 ± 2.7	1.1 ± 2.1	2.0 ± 3.2	1.4 ± 2.4
CYP3A4 inhibitors/strong affinity substrates	4.2 ± 3.8	3.4 ± 3.3	4.8 ± 4.2	4.3 ± 3.3
Simulation with iclepertin; CIB: Mean ± SD				
Control	2.3 ± 2.7	1.9 ± 2.2	2.8 ± 3.0	2.2 ± 2.3
CYP3A4 inhibitors/strong affinity substrates	5.8 ± 2.7	5.2 ± 2.2	6.3 ± 2.9	5.1 ± 2.8
LQTS				
Baseline LQTS: Mean ± SD				
Control	0.4 ± 1.3	0.2 ± 0.9	0.6 ± 1.7	0.3 ± 1.2
CYP3A4 inhibitors/strong affinity substrates	1.3 ± 2.4	0.9 ± 1.9	1.7 ± 2.6	1.1 ± 2.0
Simulation with iclepertin; LQTS: Mean ± SD				
Control	0.4 ± 1.3	0.2 ± 0.9	0.6 ± 1.7	0.3 ± 1.2
CYP3A4 inhibitors/strong affinity substrates	1.3 ± 2.4	0.9 ± 1.9	1.7 ± 2.6	1.1 ± 2.0

* Control: Individuals who did not receive any CYP3A4 inhibitors, CYP3A4 substrate inhibitors, CYP3A4 strong affinity substrates. ^¥^ Sample sizes for individuals in the control group and in the CYP3A4 perpetrator drug groups are: Total population *n* = 4,213,832 and 191,231; Commercial *n* = 1,872,277 and *n* = 65,112; Medicaid *n* = 1,888,533 and *n* = 95,443; Medicare *n* = 453,022 and *n* = 30,676, respectively. Abbreviations: CIB—CYP450 Interaction Burden; CYP450—Cytochrome P450; LQTS –Long QT Syndrome; SD—Standard Deviation.

**Table 5 pharmaceuticals-18-01453-t005:** Individuals’ characteristics with a significant increase in LQTS Score.

Individuals’ Characteristics ^¥^	Total Population	Commercial	Medicaid	Medicare
Number of individuals				
↑ LQTS units: 2 to <3	16	10	6	0
↑ LQTS units: 3 to <5	10	3	7	0
↑ LQTS units: ≥5	113	21	89	3
Number of women: n (%)	56 (45.5%)	11 (45.8%)	43 (44.8%)	2 (66.7%)
Age: mean ± SD	34.5 ± 13.6	41.0 ± 15.0	32.1 ± 11.7	59.7 ± 20.5

Abbreviations: LQTS—Long QT Syndrome; SD—Standard Deviation. ^¥^ Only individuals with an increase of ≥2 points in the LQTS score are listed in this section.

**Table 6 pharmaceuticals-18-01453-t006:** Sub-analyses performed in individuals with a recorded diagnosis of schizophrenia for their MRS, CIB, and drug-induced LQTS scores.

Groups	Total Population	Commercial	Medicaid	Medicare
Number of individuals				
Control	4,281,341	1,932,457	1,867,557	481,327
Schizophrenia *	123,722	4932	116,419	2371
MRS: mean ± SD				
Baseline				
Control:	6.2 ± 6.9	4.9 ± 5.3	7.7 ± 8.2	6.1 ± 6.1
Schizophrenia:	13.0 ± 10.3	11.3 ± 8.6	13.1 ± 10.4	11.4 ± 9.4
Simulation with iclepertin				
Control	7.1 ± 6.8	5.7 ± 5.4	8.5 ± 8.0	6.9 ± 6.0
Schizophrenia	13.6 ± 10.1	12.2 ± 8.3	13.7 ± 10.2	12.1 ± 9.2
CIB: mean ± SD				
Baseline				
Control	1.6 ± 2.7	1.1 ± 2.2	2.0 ± 3.2	1.5 ± 2.5
Schizophrenia	3.5 ± 3.9	3.1 ± 3.5	3.5 ± 3.9	3.1 ± 3.5
Simulation with iclepertin				
Control	2.4 ± 2.7	2.0 ± 2.3	2.9 ± 3.1	2.4 ± 2.5
Schizophrenia	4.2 ± 3.7	3.9 ± 3.2	4.2 ± 3.7	3.8 ± 3.3
LQTS score: mean ± SD				
Baseline				
Control	0.4 ± 1.4	0.2 ± 0.9	0.6 ± 1.7	0.4 ± 1.3
Schizophrenia	1.6 ± 2.6	1.0 ± 2.1	1.6 ± 2.6	1.6 ± 2.6
Simulation with iclepertin				
Control	0.4 ± 1.4	0.2 ± 0.9	0.6 ± 1.7	0.4 ± 1.3
Schizophrenia	1.6 ± 2.6	1.0 ± 2.1	1.6 ± 2.6	1.6 ± 2.6

* Individuals included in this group had ICD-10 codes of F20 to F29 in their medical record. Abbreviations: CIB—CYP450 Interaction Burden; LQTS—Long QT Syndrome; SD—Standard Deviation.

**Table 7 pharmaceuticals-18-01453-t007:** Model-predicted impact on medical expenditures, ED visits, and hospitalizations based on the increase in MRS following addition of iclepertin.

Groups	Total Population	Commercial	Medicaid	Medicare
Medical expenditure				
Medical expenditures (USD): mean ± SD				
Baseline	3172 ± 6068	2532 ± 5608	3349 ± 5151	5572 ± 11,512
+BI-425809	3263 ± 6142	2663 ± 5697	3411 ± 5190	5709 ± 11,704
Delta vs. Baseline	↑ 91	↑ 131	↑ 62	↑ 137
ED visits				
ED visits: mean ± SD				
Baseline	0.95 ± 1.7	0.28 ± 0.3	1.61 ± 1.4	0.49 ± 0.7
+BI-425809	0.99 ± 1.7	0.28 ± 0.3	1.64 ± 1.4	0.50 ± 0.7
Delta vs. Baseline	+0.030 ± 0.07	+0.015 ± 0.02	+0.028 ± 0.05	+0.010 ± 0.02
Hospitalizations				
Hospitalizations: mean ± SD				
Baseline	0.19 ± 1.0	0.06 ± 0.5	0.27 ± 1.0	0.19 ± 0.4
+BI-425809	0.19 ± 1.0	0.07 ± 0.5	0.28 ± 1.0	0.20 ± 0.4
Delta vs. Baseline	+0.006 ± 0.03	+0.004 ± 0.02	+0.006 ± 0.03	+0.005 ± 0.02

Abbreviations: ED—Emergency Department; SD—Standard Deviation.

**Table 8 pharmaceuticals-18-01453-t008:** Medical expenditures, ED visits, hospitalizations in individuals receiving CYP3A4 perpetrators (including CYP3A4 inhibitors and strong affinity substrates acting as competitive inhibitors towards iclepertin) ^¥^.

Groups	Total Population	Commercial	Medicaid	Medicare
Medical expenditure				
Actual medical expenditures (USD): mean ± SD				
Control: *	3028 ± 5609	2414 ± 4908	3222 ± 4841	5317 ± 10,820
CYP3A4 inhibitors/strong substrates:	6481 ± 23,197	5823 ± 15,042	6125 ± 9274	9326 ± 18,564
Simulated medical expenditures with addition of iclepertin (USD): mean ± SD				
Control: *	3116 ± 5687	2538 ± 5003	3281 ± 4884	5451 ± 11,016
Delta Simulation vs. Actual	↑ 87	↑ 124	↑ 60	↑ 135
CYP3A4 inhibitors/strong substrates:	6659 ± 12,227	6149 ± 15,091	6235 ± 9270	9499 ± 18,739
Delta Simulation vs. Actual	↑ 178	↑ 326	↑ 110	↑ 173
ED visits				
Actual ED visits: mean ± SD				
Control: *	0.91 ± 1.5	0.27 ± 0.3	1.57 ± 1.3	0.47 ± 0.07
CYP3A4 inhibitors/strong substrates:	1.87 ± 3.6	0.52 ± 0.8	2.48 ± 2.5	0.77 ± 1.1
Simulated ED visits with addition of iclepertin: mean ± SD				
Control: *				
Delta Simulation vs. Actual	+0.028 ± 0.06	+0.014 ± 0.02	+0.027 ± 0.04	+0.010 ± 0.02
CYP3A4 inhibitors/strong substrates:				
Delta Simulation vs. Actual	+0.075 ± 0.17	+0.038 ± 0.06	+0.064 ± 0.12	+0.013 ± 0.03
Hospitalizations				
Actual hospitalizations: mean ± SD				
Control: *	0.17 ± 0.9	0.06 ± 0.3	0.25 ± 0.9	0.18 ± 0.4
CYP3A4 inhibitors/strong substrates:	0.54 ± 2.5	0.21 ± 1.6	0.65 ± 2.1	0.34 ± 0.7
Simulated Hospitalizations with addition of iclepertin: mean ± SD				
Control: *				
Delta Simulation vs. Actual	+0.006 ± 0.03	+0.004 ± 0.02	+0.006 ± 0.02	+0.005 ± 0.01
CYP3A4 inhibitors/strong substrates:				
Delta Simulation vs. Actual	+0.013 ± 0.05	+0.010 ± 0.04	+0.012 ± 0.04	+0.007 ± 0.02

Abbreviations: ED—Emergency Department; SD—Standard Deviation. * Control: population who did not receive any CYP450 inhibitors, CYP3A4 substrate inhibitors, and CYP3A4 strong affinity substrates. ^¥^ Sample sizes for individuals in the control group and the CYP3A4 perpetrator drug group are: Commercial, *n* = 1,872,277 and *n* = 65,112; Medicaid, *n* = 1,888,533 and *n* = 95,443; Medicare, *n* = 453,022 and *n* = 30,676, respectively.

**Table 9 pharmaceuticals-18-01453-t009:** Sub-analyses performed in individuals with a recorded diagnosis of schizophrenia for yearly medical expenditures, yearly ED visits, and yearly hospitalizations.

Groups	Total Population	Commercial	Medicaid	Medicare
Number of individuals				
Control	4,281,341	1,932,457	1,867,557	481,327
Schizophrenia *	123,722	4932	116,419	2371
Medical expenditures: mean ± SD				
Baseline				
Control:	USD 3089 ± 5787	USD 2522 ± 5422	USD 3235 ± 4935	USD 5520 ± 11,331
Schizophrenia:	USD 6137 ± 12,145	USD 6308 ± 28,070	USD 4839 ± 7249	USD 15,968 ± 28,984
Simulation with iclepertin				
Control	USD 3180 ± 5862	USD 2652 ± 5513	USD 3297 ± 4976	USD 5656 ± 11,523
Schizophrenia	USD 6243 ± 12,220	USD 6497 ± 28,117	USD 4903 ± 7286	USD 16,277 ± 29,267
ED visits: mean ± SD				
Baseline				
Control	0.93 ± 1.6	0.27 ± 0.3	1.59 ± 1.3	0.48 ± 0.7
Schizophrenia	1.84 ± 3.5	0.61 ± 1.2	2.05 ± 2.0	1.23 ± 1.7
Simulation with iclepertin: Increase from baseline				
Control	0.030 ± 0.07	0.015 ± 0.02	0.029 ± 0.05	0.010 ± 0.02
Schizophrenia	0.036 ± 0.09	0.024 ± 0.04	0.026 ± 0.05	0.021 ± 0.05
Hospitalizations: mean ± SD				
Baseline				
Control	0.18 ± 1.0	0.06 ± 0.5	0.26 ± 0.9	0.19 ± 0.4
Schizophrenia	0.53 ± 2.4	0.36 ± 5.3	0.51 ± 1.6	0.58 ± 1.1
Simulation with iclepertin: Increase from baseline				
Control	0.006 ± 0.03	0.004 ± 0.02	0.006 ± 0.02	0.005 ± 0.02
Schizophrenia	0.010 ± 0.05	0.008 ± 0.05	0.008 ± 0.03	0.012 ± 0.03

* Individuals included in this group had ICD-10 codes of F20 to F29 in their medical record. Abbreviations: SD—Standard Deviation.

## Data Availability

The data presented in this study are available on request from the corresponding author. The data are not publicly available due to privacy.

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
