# Peer review of "Pre-Emptive Drug Safety Evaluation of Iclepertin (BI-425809) Using Real-World Data and Virtual Addition of This Medication to the Actual Drug Regimen of Individuals from Large Populations"

_pharmaceuticals, 2025, doi:10.3390/ph18101453_

Round 1
Reviewer 1 Report
Comments and Suggestions for Authors
This manuscript describes a large biosimulation of 2019 MarketScan claims in which a hypothetical iclepertin prescription (10 mg q.d.) was added to each person’s drug regimen and resulting changes in: a Medication Risk Score (MRS), CYP450 interaction burden and an LQTS score were computed. A simple pharmaco-economic translation of MRS changes to costs is also presented. The concept, using preemptive population biosimulation as a pharmacovigilance tool, is interesting and potentially useful. However, several important methodological clarifications, transparency improvements, and sensitivity/uncertainty analyses are required before this is publishable. Below are the concrete points authors should address.
- The hERG IC₅₀ is reported with inconsistent units (µM vs mM) and the Cmax / fraction-metabolized (fm) wording is confusing. Correct the IC₅₀ unit and cite the primary source (include assay system, temperature, readout and whether the IC₅₀ is for total vs unbound drug). Provide plasma protein binding (fu), compute unbound Cmax and report the unbound safety margin (IC₅₀_free ÷ Cmax_unbound). If you use an fm for CYP3A4, state how fm was estimated (in vitro intrinsic clearance scaling, clinical DDI, mass-balance etc.) and give plausible uncertainty bounds (e.g., fm = 0.8 with a plausible range 0.5–0.95). These items materially change DDI and QT predictions and must be explicit.
- The MRS algorithm is insufficiently described. Provide the full scoring algorithm (components, numeric weights, cut-points for low/medium/high), one or two worked examples (a low, medium, high risk patient showing how the score was calculated), and validation metrics (AUROC, calibration slope, or other performance vs relevant clinical outcomes used in your cost mapping). Authors need to make clear what a “unit” change in MRS represents clinically.
- Replacing formal inference with a fixed “20% rule” is not acceptable. Provide uncertainty for all key estimates: give bootstrap 95% confidence intervals (or Bayesian credible intervals) for mean MRS change, proportion shifting risk categories, LQTS score changes, and cost deltas. Describe the resampling plan (for example, 1,000 bootstrap samples stratified by age and sex), and show result distributions (histograms or violin plots) so readers can judge variability and robustness.
- The mapping from MRS changes to healthcare costs is not shown. Supply the explicit economic model (equations), sources for unit costs (ED visit, hospitalization, outpatient rates), time horizon, and whether costs are annualized or discounted. Provide one-way sensitivity analyses on key coefficients and a probabilistic sensitivity analysis (PSA) showing the distribution of cost outcomes under parameter uncertainty. Present scenarios both with and without any assumed clinical benefit of iclepertin. Include the model code or spreadsheet in Supplementary Material.
- The LQTS score derivation, point assignment and thresholds are not provided. Include the exact algorithm (variables used, scoring scheme), references validating the score for torsade/QTc prediction, and indicate what specific unit increases imply clinically (for example: “2-unit increase corresponds to X% relative risk increase” — if known). State whether sex, age or ECG history are included; if not, state this limitation and report subgroup behavior (sex, older age) separately.
- The rules for imputing missing days-supply (chronic vs acute classification and default days) are only sketched. Provide a full rule set (e.g., ATC classes or example drugs classified as chronic vs acute, default days assigned), and run sensitivity analyses re-assigning ±30/±60 days where ambiguity exists. Report how many drug records, patient-months and individual patients were affected by these imputations.
- Requiring continuous enrollment for the entire year probably biases the cohort toward more stable, possibly healthier insured persons (employer-sponsored insurance bias). Quantify and discuss this: provide baseline demographics for excluded vs included individuals if possible. If feasible, repeat primary analyses on a relaxed enrollment cohort (e.g., ≥6 months) to show sensitivity to this inclusion rule.
- All authors are company-affiliated; this is disclosed but it raises perception issues. Explicitly state the sponsor’s role in study design, data access, analysis, manuscript drafting and decision to publish.
- Although no patient identifiers are reported, the manuscript should explicitly state IRB/ethics review or exemption for use of MarketScan claims and confirm data access permissions and any data use agreements.
- minor items
- Fix unit inconsistencies and numeric typos throughout (consistent use of nM, µM etc).
- List software, packages and versions used (Python, SciPy, statsmodels, SQL etc).
- Clarify language around “not significant” — specify statistical vs clinical meaning and report CIs.
- Provide clearer subgroup denominators (e.g., schizophrenia cohort) and basic stratified summaries (age, sex, polypharmacy).
Author Response
Response to Reviewer #1
We thank this reviewer for a very insightful review of our paper. Unfortunately, some analyses requested could not be performed as it would require months of data processing. Furthermore, due to acquisition of Tabula Rasa HealthCare, Inc. (TRHC) by independent corporations, access to primary data and algorithms used at the time of analyses are no longer available.
Comment 1:
The hERG IC₅₀ is reported with inconsistent units (µM vs mM) and the Cmax / fraction-metabolized (fm) wording is confusing. Correct the IC₅₀ unit and cite the primary source (include assay system, temperature, readout and whether the IC₅₀ is for total vs unbound drug). Provide plasma protein binding (fu), compute unbound Cmax and report the unbound safety margin (IC₅₀_free ÷ Cmax_unbound). If you use an fm for CYP3A4, state how fm was estimated (in vitro intrinsic clearance scaling, clinical DDI, mass-balance etc.) and give plausible uncertainty bounds (e.g., fm = 0.8 with a plausible range 0.5–0.95). These items materially change DDI and QT predictions and must be explicit.
Response 1:
Units associated with hERG IC₅₀ have been corrected and reported as mM throughout the text.
“hERG IC50 was determined by patch-clamp in HEK293 cells as described previously [26]” has been added to the text.
Plasma protein binding (fu) is 19.6% so that the unbound safety margin calculated as IC₅₀ free / Cmax unbound is 689 and is not expected to cause major hERG block. Some organizations have proposed calculating this ratio to assess potency for drug block of hERG. Although this could provide some indications, our group has published numerous studies and identified potent hERG blockers without considering the safety ratio as a “conditional” factor, as binding to hERG channel protein is from the inside of the cardiac myocyte plasma membrane, not from the outside. So, conceptually, the use of the proposed safety ratio is flawed. The free intracellular concentration of a drug is what dictates the risk of hERG protein block by a drug, not its extracellular concentration. Obviously, there are driving forces that regulate the intracellular concentration of a drug such as its extracellular free concentration. But several other elements, including its intracellular metabolism in cardiac cells (by CYP450s such as CYP2J2), its transport by influx and efflux transporters (ABCG2 is highly expressed in cardiac myocytes as is SLCO2B1), and its binding to intracellular proteins are major elements dictating the intracellular (myocyte concentration) and binding to and blocking of hERG protein.
The fraction metabolized by CYP3A4 was derived from a clinical DDI trial with Itraconazole (Iclepertin is not P-gp substrate). The increase in AUC0-inf was 5.97-fold (90% CI range 5.14-6.93); based on these changes, the CL (3A4 related) was reduced to 16.8% (90% CI range 14-19%) and hence fm would be 0.832 (range 0.81 – 0.86).
Comment 2:
The MRS algorithm is insufficiently described. Provide the full scoring algorithm (components, numeric weights, cut-points for low/medium/high), one or two worked examples (a low, medium, high risk patient showing how the score was calculated), and validation metrics (AUROC, calibration slope, or other performance vs relevant clinical outcomes used in your cost mapping). Authors need to make clear what a “unit” change in MRS represents clinically.
Response 2:
The MRS factors included in its calculation have been described in detail in patent #11361856: Population-based medication risk stratification and personalized risk score. This detailed publication of the invention (>200 pages) relates to a system and method for both population-based medication risk stratification and for generating a personalized medication risk score. The system and method may pertain to a software that relates pharmacological characteristics of medications and patient's drug regimen data into algorithms that (1) enable identification and/or prognosis of high-risk patients for adverse drug events within a population distribution, and (2) allow computation of a personalized medication risk score which provides personalized, evidence-based information for safer drug use to mitigate medication risks. We have added this reference to the medication risk score description.
This risk score has been used to demonstrate the relationship between a medication risk score and patients’ outcomes in large populations (220,000 patients; Michaud et al., Association of the MedWise Risk Score with health care outcomes. Am J Manag Care. 2021 Sept 27; (16 Suppl): S280-S291) and independently by the DARTNet Institute in a retrospective analysis of 427,103 patients (Ratigan et al., Longitudinal Association of a Medication Risk Score with Mortality Among Ambulatory Patients Acquired Through Electronic Health Record Data. Journal of Patient Safety. 2021;17:249-255). It has also been used in numerous publications relating patients’ outcomes to inappropriate drug regimen.
Comment 3:
Replacing formal inference with a fixed “20% rule” is not acceptable. Provide uncertainty for all key estimates: give bootstrap 95% confidence intervals (or Bayesian credible intervals) for mean MRS change, proportion shifting risk categories, LQTS score changes, and cost deltas. Describe the resampling plan (for example, 1,000 bootstrap samples stratified by age and sex), and show result distributions (histograms or violin plots) so readers can judge variability and robustness.
Response 3:
We thank the reviewer for pointing out this potential issue. The paragraph in question (lines 407-412) addressed this, but it could be clarified.
The purpose of confidence intervals, p-values, etc. in this context would typically be to compare the difference between intervention and control groups against a null hypothesis of "no effect." This is to establish how credible the null hypothesis is based on the observed data: what is the probability that the difference between the 2 groups is due to chance in the randomization process? That question does not apply to this study.
This study involves a deterministic simulation, no randomization. Exactly one fictitious claim is added to all patients’ drug regimens, which are run through the deterministic MRS algorithm. Therefore, there is no null hypothesis to compare against. The probability that any difference is due to chance is, by definition, 0. It is meaningless to check for statistical significance in this case.
Furthermore, all the analyses after generating the MRS score are likewise deterministic (including model predictions).
Therefore, the key question in this study is not statistical significance (a signifier that the change in score/outcomes might be "real") but of clinical significance (is the change in score/outcomes clinically meaningful and worth the expected benefits). The 20% rule was adopted as a check of clinical significance.
To clarify these issues, we propose replacing the paragraph in lines 407-412 as follows: Given the deterministic nature of this study, statistical significance analysis is not warranted. Exactly one fictitious claim was added to all patients’ drug regimens, which were analyzed using the deterministic MRS algorithm. Therefore, there is no null hypothesis to compare against. Common sense is to be used to determine if differences observed are meaningful and clinically relevant [24]. For factors associated with changes in the MRS and its components, a difference of at least 20% was considered clinically relevant.
Comment 4:
The mapping from MRS changes to healthcare costs is not shown. Supply the explicit economic model (equations), sources for unit costs (ED visit, hospitalization, outpatient rates), time horizon, and whether costs are annualized or discounted. Provide one-way sensitivity analyses on key coefficients and a probabilistic sensitivity analysis (PSA) showing the distribution of cost outcomes under parameter uncertainty. Present scenarios both with and without any assumed clinical benefit of iclepertin. Include the model code or spreadsheet in Supplementary Material.
Response 4:
The association between the Medication Risk Score used in this study and patients’ outcomes have been extensively described in Michaud et al., Association of the MedWise Risk Score with health care outcomes. Am J Manag Care. 2021 Sept 27; (16 Suppl): S280-S291), Ratigan et al., Longitudinal Association of a Medication Risk Score with Mortality Among Ambulatory Patients Acquired Through Electronic Health Record Data. Journal of Patient Safety. 2021;17:249-255 and Bankes et al., Association of a Novel Medication Risk Score with Adverse Drug Events and Other Pertinent Outcomes Among Participants of the Program of All-Inclusive Care for the Elderly. Pharmacy 2020;8:87 doi:10.3390/pharmacy8020087.
Comment 5:
The LQTS score derivation, point assignment and thresholds are not provided. Include the exact algorithm (variables used, scoring scheme), references validating the score for torsade/QTc prediction, and indicate what specific unit increases imply clinically (for example: “2-unit increase corresponds to X% relative risk increase” — if known). State whether sex, age or ECG history are included; if not, state this limitation and report subgroup behavior (sex, older age) separately.
Response 5:
The Long QT Score calculation has been described in detail in Patent #10890577: Treatment methods having reduced drug-related toxicity and methods of identifying the likelihood of patient harm for prescribed medications (cited in the manuscript). This publication describes methods of determining whether specific drugs or patients carry an increased risk of causing or developing, respectively, long QT syndrome or Torsades de Pointes. and methods of treating such patients. This information has been added to the Methods section, under the Medication Risk Score heading. Such method has been used and further described in Michaud V, Dow P, Al Rihani SB, Deodhar M, Arwood M, Cicali B, Turgeon J. Risk assessment of drug-induced Long QT Syndrome for some COVID-19 repurposed drugs. Clin Transl Sci 2020;14(1):20-28 (cited). The methodology has also been presented at the American Heart Association meeting in 2016: Steffen, L.E., Knowlton, C.H., Turgeon, J. Development of a drug-specific Long QT-JT Index for prediction of drug-induced QT Prolongation. American Heart Association Annual Meeting Scientific Sessions. Circulation 2016;134:A15939 and in Steffen, L.E., Knowlton, C.H., Turgeon, J. Risk Identification of drug-induced Long QT Syndrome using a Patient-Specific Long QT-JT Risk Score. American Heart Association Annual Meeting Scientific Sessions. Circualtion;134:A16114
Comment 6:
The rules for imputing missing days-supply (chronic vs acute classification and default days) are only sketched. Provide a full rule set (e.g., ATC classes or example drugs classified as chronic vs acute, default days assigned), and run sensitivity analyses re-assigning ±30/±60 days where ambiguity exists. Report how many drug records, patient-months and individual patients were affected by these imputations.
Response 6:
Example drugs and default days assigned were provided in lines 368-372: If drugs were considered as chronic medication (e.g., opioids, antihypertensive agents), the drug was considered continuous for the full year (from this date) for the risk stratification analysis. If the medications were considered acute (e.g., antibiotic, seasonal allergy medications), the drug was considered administered for 30 days (from the date prescribed). We added more examples for chronic medications.
We do not consider that providing a full set of ATC drugs (as of now there are more than 14,000 different ATC numbers) would change anything in the conclusion of our analyses. Chronic vs acute drug characteristics were manually determined by our pharmacists (co-authors on the publication). For claims with missing “day supply” data, these three pharmacists reviewed each claim, determined which drugs were mostly likely prescribed for acute conditions vs chronic conditions (e.g. antibiotics, antivirals, drugs for seasonal allergy compared to antihypertensive, statins, etc.). Assumptions were made which could have introduce a possibility for “erroneous assignment”. The objective was not to overinflate the MRS and consider “chronic” drugs that are usually prescribed for a short duration. The “error” is consistent for all claims with missing data.
The goal was not to have a “perfect real time” drug regimen, but realistic patterns of drug regimens taken.
Comment 7:
Requiring continuous enrollment for the entire year probably biases the cohort toward more stable, possibly healthier insured persons (employer-sponsored insurance bias). Quantify and discuss this: provide baseline demographics for excluded vs included individuals if possible. If feasible, repeat primary analyses on a relaxed enrollment cohort (e.g., ≥6 months) to show sensitivity to this inclusion rule.
Response 7:
We agree that this could represent a bias if we were to conduct prospective analyses over a cohort of health plan members during a specific year. However, this is not the case in this manuscript, as data included (from the Merative database) comes from various and multiple sources. We do not consider that this criterion created any bias, as exposure time to the various drugs was not considered in our model. Our objective was to assess stable drug regimen under chronic treatments, not acute drug-to-drug interactions. It could be relevant if we were trying to establish whether longer exposure to a drug could increase the risk of associated side-effects, but this is not the purpose of this study.
Comment 8:
All authors are company-affiliated; this is disclosed but it raises perception issues. Explicitly state the sponsor’s role in study design, data access, analysis, manuscript drafting and decision to publish.
Response 8:
We fully recognized that all authors are currently part of the industry, and we have disclosed such potential conflict of interest. The analyses were performed by GalenusRx scientists (who were employed by Tabula Rasa at that time) without the intervention of Boehringer Ingelheim scientists. This publication does not promote dug efficacy but rather openly assesses drug safety. Although a drug candidate was used for the analyses, the objective of the publication is to demonstrate that novel approaches could be used to assess drug safety, rather than waiting for side-effects to manifest themselves in patients.
Comment 9:
Although no patient identifiers are reported, the manuscript should explicitly state IRB/ethics review or exemption for use of MarketScan claims and confirm data access permissions and any data use agreements.
Response 9:
This study includes analyses performed on an existing, de-identified dataset, and therefore meets exemption under 45 CFR 46.104(d)(4).
Minor items
- Fix unit inconsistencies and numeric typos throughout (consistent use of nM, µM etc).
This has been done for hERG IC50
- List software, packages and versions used (Python, SciPy, statsmodels, SQL etc).
Software and packages used were mentioned in lines 404-405. Specific versions cannot be provided, since the original environments cannot be recovered due to job changes. However, these are fairly mature, stable packages.
- Clarify language around “not significant” — specify statistical vs clinical meaning and report CIs.
We meant clinically significant, as described above under Comment #3
- Provide clearer subgroup denominators (e.g., schizophrenia cohort) and basic stratified summaries (age, sex, polypharmacy).
For schizophrenia, we have used specific ICD10 codes F-20 to F-29 as described under lines 183 and 373, as well as in the footnotes of Tables 6 and 9. Table 1 summarizes the characteristics of the population for age, gender, and number of medications.
Reviewer 2 Report
Comments and Suggestions for Authors
Although biosimulation is not my primary area of expertise, I have carefully reviewed the manuscript and provided constructive feedback to the best of my ability. My comments on the study by Sebastian Härtter et al., entitled 'Pre-emptive drug safety evaluation of iclepertin (BI-425809) using real-world data and virtual addition of this medication to the actual drug regimen of individuals from large populations', are below. While the study addresses a relevant and timely topic, some points should be clarified or revised to improve its clarity and scientific rigour.
Major and specific comments:
1. Lines 363–365: The number of individuals reported in the Methods section should be removed, as this information belongs in the Results section.
2. Methods section: The definitions of 'commercially insured', 'eligible for Medicare' and 'eligible for Medicaid' should be clarified. Providing the exact criteria would help international readers to better understand the population characteristics.
3. Line 375: A clearer and more detailed description of the Medication Risk Score is needed, including how it is calculated and interpreted.
4. Language: The manuscript would benefit from careful editing to improve the fluency, consistency and accuracy of the English.
5. Figures: To facilitate interpretation, it would be helpful if each figure were accompanied by a more detailed description.
6. Discussion: Please briefly address the potential limitations of using real-world data for biosimulation, such as variability in data quality and incomplete information.
7. Discussion: It would also be useful to discuss the methodological limitations of biosimulation itself. For instance, consider the assumptions underlying the models and their potential impact on the results.
Author Response
Response to Reviewer #2
We thank this reviewer for their review of our paper. We have thoroughly responded to comments.
Comment 1:
Lines 363–365: The number of individuals reported in the Methods section should be removed, as this information belongs in the Results section.
Response 1:
This information has been moved to the Result section as required.
Comment 2:
Methods section: The definitions of 'commercially insured', 'eligible for Medicare' and 'eligible for Medicaid' should be clarified. Providing the exact criteria would help international readers to better understand the population characteristics.
Response 2:
Commercially insured are individuals obtaining insurance either through the employer or through a family plan. Medicaid is a wide-ranging, federal, health care program mostly for low-income individuals of any age. Medicare is also federal program for health care coverage of individuals 65 years old and above. This information has been added at the beginning of the Method Section.
Comment 3:
Line 375: A clearer and more detailed description of the Medication Risk Score is needed, including how it is calculated and interpreted.
Response 3:
The MRS factors included in its calculation have been described in detail in patent #11361856: Population-based medication risk stratification and personalized risk score. This detailed publication of the invention (>200 pages) relates to a system and method for population-based medication risk stratification and for generating a personalized medication risk score. The system and method may pertain to a software that relates pharmacological characteristics of medications and patient's drug regimen data into algorithms that (1) enable identification and/or prognosis of high-risk patients for adverse drug events within a population distribution, and (2) allow computation of a personalized medication risk score which provides personalized, evidence-based information for safer drug use to mitigate medication risks. We have added this reference to the medication risk score description.
This risk score has been used to demonstrate the relationship between a medication risk score and patients’ outcomes in large populations (220,000 patients; Michaud et al., Association of the MedWise Risk Score with health care outcomes. Am J Manag Care. 2021 Sept 27; (16 Suppl): S280-S291) and independently by the DARTNet Institute in a retrospective analysis of 427,103 patients (Ratigan et al., Longitudinal Association of a Medication Risk Score with Mortality Among Ambulatory Patients Acquired Through Electronic Health Record Data. Journal of Patient Safety. 2021;17:249-255). It has also been used in numerous publications relating patients’ outcomes to inappropriate drug regimen.
Comment 4:
Language: The manuscript would benefit from careful editing to improve the fluency, consistency and accuracy of the English.
Response 4:
The manuscript has been reviewed by our publication writer for clarity.
Comment 5:
Figures: To facilitate interpretation, it would be helpful if each figure were accompanied by a more detailed description.
Response 5:
Additional information has been added to the Figure 1 legend.
Although changes in MRS distribution were observed for each group, the extent of changes was of relatively minor, suggesting a good safety profile of iclepertin.
Comment 6:
Discussion: Please briefly address the potential limitations of using real-world data for biosimulation, such as variability in data quality and incomplete information.
Response 6:
Comments about the limitations of the approach presented have been added to the Discussion.
Further, using real-world data presents several advantages, including the acquisition and analysis of data on many individuals in a short time. However, real world data often requires significant clean-up, is static, and is not aways uniform between all individuals, which can be a limitation when needing specific inclusion criteria to perform analyses.
Comment 7:
Discussion: It would also be useful to discuss the methodological limitations of biosimulation itself. For instance, consider the assumptions underlying the models and their potential impact on the results.
Response 7:
Comments about the limitations of the approach presented have been added to the Discussion.
As mentioned previously, relevant side-effect frequency for iclepertin is absent from the FAERS, and therefore, our approach did not estimate benefits associated with iclepertin or consider other potential side-effects. This is a significant limitation as only 2 out of 5 factors included in the MRS could be assed.
Round 2
Reviewer 1 Report
Comments and Suggestions for Authors/